# Association between Milk Electrical Conductivity Biomarkers with Lameness in Dairy Cows

**DOI:** 10.3390/vetsci10010047

**Published:** 2023-01-09

**Authors:** Algimantas Paulauskas, Vida Juozaitienė, Karina Džermeikaitė, Dovilė Bačėninaitė, Gediminas Urbonavičius, Saulius Tušas, Evaldas Šlyžius, Walter Baumgartner, Arūnas Rutkauskas, Ramūnas Antanaitis

**Affiliations:** 1Department of Biology, Faculty of Natural Sciences, Vytautas Magnus University, K. Donelaičio 58, LT-44248 Kaunas, Lithuania; 2Large Animal Clinic, Veterinary Academy, Lithuanian University of Health Sciences, Tilžės Str. 18, LT-47181 Kaunas, Lithuania; 3Department of Animal Breeding, Veterinary Academy Lithuanian University of Health Sciences, Tilžės Str. 18, LT-47181 Kaunas, Lithuania; 4Institute of Animal Rearing Technologies, Veterinary Academy Lithuanian University of Health Sciences, Tilžės Str. 18, LT-47181 Kaunas, Lithuania; 5University Clinic for Ruminants, University of Veterinary Medicine, Veterinaerplatz 1, A-1210 Vienna, Austria

**Keywords:** milk to electrical conductivity, locomotion score, cortisol, stress

## Abstract

**Simple Summary:**

The aim of this study was to determine how lameness in cows correlates with blood cortisol levels and milk electrical conductivity, and which of the indicators of electrical conductivity during milk flow phases can be used to predict the early risk of cow lameness. During this study, we found that the average cortisol concentration in the blood is associated with the laminitis score and milk electrical conductivity indicators. Cows with a higher score of lameness had higher cortisol concentrations and higher milk conductivity. We recommend repeating this investigation using current methodology in a large number of dairy herds because we could not draw any broad conclusions about the cause and effects because we only investigated the correlation between lameness and milk conductivity in one herd.

**Abstract:**

Early identification of lameness at all phases of lactation improves milk yield and reduces the incidence of mastitis in the herd. According to the literature we hypothesized that there are associations of electrical conductivity variables of milk flow with lameness in dairy cows. The aim of this study was to determine if blood cortisol and electrical conductivity in the milk flow phases correlate with each other and whether they are related to cow lameness. On one farm, out of 1500 cows, 64 cows with signs of lameness and 56 healthy cows were selected with an average of 2.8 lactations and 60 days in the postpartum period. A local veterinarian who specializes in hoof care treatments identified and scored lameness. During evening milking, the milk flow of all 120 cows was measured using electronic milk flow meters (Lactocorder^®^, WMB AG, Balgache, Switzerland). Before each milking, two electronic mobile milk flow meters (Lactocorders) were mounted between the milking apparatus and the milking tube to take measurements. We found that the average cortisol concentration in the blood of the studied cows was significantly correlated with the laminitis score. Results of this study indicate that the number of non-lame cows with a milk electrical conductivity level of <6 mS/cm even reached 90.8–92.3% of animals. Milk electrical conductivity indicators ≥ 6 mS/cm were determined in 17.8–29.0% more animals in the group of lame cows compared to the group of non-lame cows. According to our study, we detected that blood cortisol concentration had the strongest positive correlation with milk electrical conductivity indicators. Cows with a greater lameness score had a higher cortisol content and milk conductivity.

## 1. Introduction

Dairy cow lameness is one of many factors that have a detrimental impact on a dairy cow’s health and wellbeing [1]. The prevalence of clinical lameness in cows ranges from 26 to 54% [2,3,4,5,6]. However, herd management appear to be grossly underestimating it [7]. Lameness can be measured reproducibly [8] and can be identified as the most representative animal-based welfare indicator in dairy cattle [9]. Milk producers’ costs rise as milk production falls and the calving interval of lame dairy cows increases [10]. Lameness in cows starts a negative chain reaction that has various consequences for both the farmer and the cow, including lower milk yield (a reduction of around 20%), loss of reproduction, lack of weight increase, and, in some cases, animal culling. According to studies, lower milk supply occurs prior to therapy for infectious lameness but varies depending on the type of lesion present [11]. Furthermore, the decrease in milk output is larger in older cows and cows with severe lameness [12]. Severe lesions, which are less prevalent, can cause three times the economic losses as moderate lesions [13]. Dairy producers face numerous obstacles and opportunities as the usage of automated milking systems (AMSs) grows. The use of AMSs has the added benefit of monitoring cow-level milking frequency, quarter-level production, and milk quality, which can help diagnose sickness [14]. As a result of cow lameness, some indicators recognized by automated milking systems change [15]. However, not all health issues can be detected electronically, and producers must still physically inspect and gather cows for milking if the interval between milkings is too long [8]. To detect lameness in fresh dairy cows, milk yield, milk electrical conductivity, milk composition (protein, lactose, and fat), walking activity, rumination length, and pH value in the reticulorumen will be used. We recommend that the trial period in future research be extended to 30 days or more before clinical signs of lameness appear. 

In dairy cattle, pain from lameness may act as a stressor [16]. Adverse events cause adrenal reactions, which results in a rise in glucocorticoid levels [17]. Cortisol has been used in lame cattle as a stress biomarker [17]. Serum cortisol levels are increased in cows diagnosed with lameness on the day of diagnosis [18] Clinically (i.e., laminitis, metritis, mastitis) and physiologically affected (parturition) cows had higher cortisol levels than clinically healthy cows [19]. Reduced cortisol secretion with extended exposure to a stressor, such as chronic lameness, may indicate a mechanism to minimize prolonged exposure to elevated cortisol concentrations [20]. Despite the unquestionable importance of lameness (caused by claw horn abnormalities), little is known about the genesis and pathophysiology of lameness-associated noninfectious diseases [4]

Previous research has found that AMS visits, cow productivity, and milking intervals are all affected by lameness. The sum of these negative outcomes has a significant influence on herd profitability as well as cow health and well-being. It is strongly recommended that AMS factors be thoroughly investigated in order to ensure proper management of dairy cow performance and hoof health [21]. According to Miguel-Pacheco et al. [22], more research is needed to investigate the possible applications and benefits of the technologies available in AMS as a tool for evaluating and monitoring cow health status. Lameness can be recognized 7–10 days before clinical signs occur using electronic devices that record cow walking time, which is related with decreased activity in cows [23]. Cows with lameness eat less and are less active than non-lame cows [24,25]. 

Automated lameness detection could fill a knowledge limitation by providing relevant cow and herd information, particularly for mild and moderately lame cows. Early identification and automatic drafting could minimize the time between the start of lameness and treatment, preventing cases from becoming severe, hastening recovery, increasing output, and enhancing welfare [26]. Precision livestock farming is acknowledged as essential for future dairy producers since it allows for constant monitoring of animal health and welfare throughout production [27]. 

Unfortunately, lameness is not always easy to detect. According to Espejo et al. (2006), farm managers recognized only 30% of the lameness instances reported by a qualified observer [7]. According to Garvey [28], controlling infectious illness in dairy herds requires early detection and prevention. Early identification and treatment of lameness at all phases of lactation improves milk yield and reduces the incidence of mastitis in the herd. According to the literature, we hypothesized that there are associations of electrical conductivity variables of milk flow and blood cortisol concentration with lameness in dairy cows. The objective of this study was to find out how lameness in cows correlates with blood cortisol levels and milk electrical conductivity and the number of somatic cells in milk. 

## 2. Methods

### 2.1. Animal and Experimental Design

This transversal study was conducted on a dairy farm in Lithuania. The cows on the farm are milked twice a day by 24 parallel milking parlors (DeLaval VMS; DeLaval International AB Tumba, Botkyrka, Sweden). The milking capabilities of cows were evaluated twice, during the evening and morning milking. The cows in this lactation study did not receive veterinary treatment, and correct hoof trimming was not performed at least four weeks prior to the study.

Diets were designed in compliance with the National Research Council’s standards [29]. The energy requirements of 550 kg lactating Holstein dairy cows producing 35 kg/day were met or exceeded by the diets. TMR made up of 35% corn silage, 10% grass silage, 5% grass hay, and 50% grain concentrate mash for cows (50 percent barley and 50 percent wheat). NFC (percentage of DM)—38.7; CP (percentage of DM)—15.8; NEL 84 (Mcal/kg)—1.6. TMR was administered to the cows twice a day, at 10:00 a.m. and 8:00 p.m.

The selection criteria were identified from a farm with 1500 Lithuanian black and white dairy cows. Cows with two or more lactations met the inclusion requirements. 64 cows with signs of lameness and 56 healthy cows were selected with an average of 2.8 lactations and 60 days of the postpartum period. According to the conventional technique described, lameness was detected by a local veterinarian who specialized in hoof care procedures described by Sprecher et al. [8]: 1 = normal (*n* = 56), 2 = presence of a slightly asymmetric gait (*n* = 24), 3 = the cow clearly favored one or more limbs (moderately lame) (*n* = 33), 4 = severely lame (*n* = 7). For four weeks, the same observer rated visual locomotion once every week.

### 2.2. Measurements

The milk flow of 120 cows was measured using electronic milk flow meters (Lactocorder^®^, WMB AG, Balgache, Switzerland) during evening milking. Before each milking, two electronic mobile milk flow meters (Lactocorders) were mounted between the milking apparatus and the milking tube to take measurements. LactoPro 5.2.0 software was used to analyze milk flow data (Biomelktechnik Swiss). The first table describes the milk flow traits of the cows studied in this experiment (Table 1). 

During the same general clinical examination, blood samples were taken from the coccygeal veins of cows that had been fixed with appropriate equipment using a tube devoid of anti-coagulant (BD Vacutainer, Crawley, UK) and centrifuged for 10–15 min at 3500 RPM, 20 °C. The samples were delivered to the Large Animal Clinic of the Lithuanian University of Health Sciences’ Laboratory of Clinical Tests. TOSOH^®^ AIA-360 (South San Francisco, CA, USA) automated analyzer was utilized to measure cortisol levels, which uses a competitive fluorescence enzyme immunoassay that runs in compact, single-use test cups that contain all necessary reagents. Previously, data for human and canine T4 and cortisol accuracy and performance, including analyte recovery and dilutional testing, had been evaluated [30]. The daily inspections, calibration curves, and maintenance procedures were carried out in accordance with the System Operator’s Manual.

The number of somatic cells in milk was determined by flow cytometry using the Somascope CA-3A4 (Delta Instruments, Drachten, The Netherlands) at the State Laboratory for Milk Control.

### 2.3. Treatment of Lameness

Naxcel (100 mg ceftiofur/mL; Zoetis Canada, Kirkland, QC, USA) was given subcutaneously at a dose of 2.2 mg/kg body weight to all lame cows. Every 24 h, the procedure was repeated. Simultaneously, Rimadyl Cattle^®^ solution (50 mg carprofen/mL; Zoetis, Belgium) was given subcutaneously once at 1.4 mg per 1 kg body weight.

### 2.4. Statistical Analysis

Statistical analysis was performed using the IBM SPSS Statistics software (version 25.0, IBM, Munich, Germany). The mean standard error of the mean of a sample (M SEM) is offered for analysis of the normally distributed data of blood cortisol and milk electrical conductivity (as determined by the Shapiro-Wilk normality test). The significance of the association between blood cortisol classes and milk electrical conductivity measures was determined using Pearson’s chi-square test of independence. Pearson’s paired linear dependence was used to compute the link between the electrical conductivity of milk and the concentration of cortisol in cow blood, and the Spearman coefficient was used to examine the relationship between the electrical conductivity of milk and cow lameness scores. 

Using a binary multivariable logistic regression, we investigated the relationship between lameness, blood cortisol levels, milk electrical conductivity and the number of somatic cells in milk. According to the electrical conductivity of milk, cows were divided into two groups: <6 mS/cm and ≥6 mS/cm, blood cortisol: <1.00 µg/dL and ≥1.00 µg/dL, and the number of somatic cells in milk: <200 thousand/mL and ≥200 thousand/mL. On the basis of the sample mode, the regression model explanatory variables were separated into two category classes. The logistic regression results are shown as odds ratios (OR) and 95% confidence intervals (PI).

## 3. Results

The average cortisol concentration in the blood of the studied cows (M = 1.04 ± 0.064 µg/dL) was significantly correlated with the laminitis score (r = 0.630, *p* < 0.001). In the non-lame cow group (Figure 1), animals with blood cortisol <1.00 µg/dL accounted for 67.3% of the cows, while only 25.9% of the animals in the lame cow group (*p* < 0.001).

ELHMF—Electrical conductivity at the maximum milk flow (mS/cm); ELAP—Electrical conductivity during the first few minutes of milking (mS/cm) (beginning peak level of the electrical conductivity); ELMAX—Maximum electrical conductivity after reaching the highest milking speed (mS/cm); ELMNG—Maximum electrical conductivity following main milking (mS/cm); ELAD—Beginning of the electrical conductivity peak difference (mS/cm).

The number of non-lame cows with a milk electrical conductivity level of <6 mS/cm even reached 90.8–92.3% of animals in terms of ELHMF and ELAD and ELMNG and 55.4 and 75.4% in terms of ELAP and ELMAX indicators, respectively. Milk electrical conductivity indicators ≥ 6 mS/cm were determined in 17.8–29.0% more animals in the group of lame cows compared to the group of non-lame cows (Table 2). 

Blood cortisol concentration had the strongest positive correlation with milk electrical conductivity indicators such as ELHMF, ELMAX and ELAP (r = 0.51–0.704, *p* < 0.001), and lameness score—with ELHMF and ELAP (r = 0.339 and r = 0.404, respectively; *p* < 0.001) (Table 3).

ELHMF—Electrical conductivity at the maximum milk flow (mS/cm); ELAP—Electrical conductivity during the first few minutes of milking (mS/cm) (beginning peak level of the electrical conductivity); ELMAX—Maximum electrical conductivity after reaching the highest milking speed (mS/cm); ELMNG—Maximum electrical conductivity following main milking (mS/cm); ELAD—Beginning of the electrical conductivity peak difference (mS/cm). r—correlation coeficient, *p*—statistical significance of the correlation coefficient. * *p* < 0.01.

Analysis of the final statistical model of the studied influencing factors on lameness in cows showed that there is a significant association between an increase in milk ELHMF ≥ 6 mS/cm (6.074 times, *p* < 0.001) and cortisol concentration in blood above 1.00 µg/dl (5.704 times, *p* < 0.001). Logistic regression analysis showed that of all milk electrical conductivity indicators, only ELHMF was significantly associated with the occurrence of cow lameness (Table 4).

ELHMF—Electrical conductivity at the maximum milk flow. ELHMF classes: class 0 < 6 mS/cm and class 1 ≥ 6 mS/cm; blood cortisol classes: class 0 < 1.00 µg/dL and class 1 ≥ 1.00 µg/dL. *p—p*-value (statistically significant with a *p*-value < 0.05); OR—odds ratio, CI—95% confidence interval.

The number of cows with milk ELHMF < 6 mS/cm in the group of healthy animals was only 6.9% of animals and increased (from 11.0% to 91.9%) with an increase in lameness score from 1 to 4 (*p* < 0.001) (Figure 2).

Reverse logistic analysis, which showed that a one-point increase in lameness intensity was associated with a 2.642-fold increase in the risk of ELHMF values exceeding 6 mS/cm (*p* < 0.001).

On the other hand, in the group of lame cows, we found 76.56% of milk samples with the level of somatic cells ≥ 200 thousand/mL, and in the group of non-lame cows, only 28.57% of such samples. (χ^2^ = 27.707, *p* < 0.001). The intensity of lameness was associated with an increase in the number of cows with the indicated level of somatic cells in milk and a decrease in the number of cows in which the level of somatic cells in milk samples was <200 thousand/mL (χ^2^ = 30.269, *p* < 0.001) (Figure 3).

In the group of cows with ELHMF of milk < 6 mS/cm, 36.07% of milk samples with the level of somatic cells ≥ 200 thousand/mL were detected, and in the group of cows with ELHMF ≥ 6 mS/cm—even 72.88. % of such samples were detected (χ^2^ = 16.374, *p* < 0.001).

To analyze the factors associated with lameness in cows, the multivariable logistic regression model (presented in Table 4) was supplemented with somatic cell levels in milk. Applying a backward stepwise logistic regression model, eliminating all non-significant explanatory variables (according to the significance of the Wald criterion), no significant effects of somatic milk cells was found. This can be explained by the fact that highly correlated variables cannot be used in multiple logistic regression models to ensure the absence of multicollinearity, and as mentioned earlier, somatic cell count in milk was significantly associated with ELHMF (*p* < 0.001). 

## 4. Discussion

Lameness is a major welfare concern in dairy cattle [31], causing pain [32], decreased milk production [33], decreased longevity [34] and decreased reproductive function [35]. 

We found that the average cortisol concentration in the blood of the studied cows was significantly correlated with the laminitis score. Existing data on the influence of lameness on cortisol levels in cow is inconclusive. For example, O’Driscoll et al. [18] found that cows suffering lameness from sole ulcers had greater blood cortisol levels than healthy cows. On the other hand, Almeida et al. [36] and Walker et al. [37] stated that despite numerical differences (up to 43% higher values in lame animals), there was no statistically significant difference in cortisol levels in blood or milk between lame and sound cows). The ability to detect a distinct increase in milk cortisol in lame animals may be related to the cause and duration of the distress. In this regard, Almeida et al. [36] and O’Driscoll [18] advocated using the cortisol-to-dehydroepiandrosterone (DHEA) ratio as a marker of inflammation. In dairy cattle, pain from lameness may act as a stressor [16]. Adverse events cause adrenal reactions, which result in an increase in glucocorticoid concentration [17]. Cortisol has been used in lame cattle as a stress biomarker [38]. Serum cortisol levels are greater in cows with lameness on the day of diagnosis [18]. Cortisol levels were higher in clinically compromised (laminitis, metritis, mastitis) and physiologically compromised (parturition) cows than in clinically healthy cows [19]. Comin et al. [19] used data from the on-farm computer to categorize the animals, therefore it is possible that the personnel-based diagnosis contained more acute and severe lame cows. O’Driscoll et al. [18] discovered that cortisol levels in cows with sole ulcers were increased on the day of diagnosis. According to some studies the cortisol level in cows suffering from sole hemorrhages were also increased [18]. The transfer from acute to chronic pain stimuli is an adaptive coping mechanism in the body that allows cortisol levels to recover to normal [39]. 

Results of this study indicate that the number of non-lame cows with a milk electrical conductivity level of <6 mS/cm even reached 90.8–92.3% of animals in terms of ELHMF, ELAD and ELMNG as well as 55.4 and 75.4% in terms of ELAP and ELMAX indicators, respectively. Milk electrical conductivity indicators ≥ 6 mS/cm were determined in 17.8–29.0% more animals in the group of lame cows compared to the group of non-lame cows.

Monitoring affected milk electrical conductivity has demonstrated value as an indirect and rapid method of detecting subclinical mastitis [40]. The technology works by measuring potassium, sodium, and other free ions, particularly chloride (Cl), which is proportional to electrical conductivity. Normal milk contains 75–130 mg/100 mL of Cl; however, inflammation can cause the amount of Cl to rise to 111–198 mg/100 mL. Depending on the type of mastitis developing, these alterations might occur quickly and randomly [40]. Lameness has a substantial impact on animal health, production, welfare, and reproduction in crossbred dairy cattle herds [41]. In lame cows, severe discomfort changes their usual rising and lying behavior. Mastitis risk increased as cows lay down for longer periods of time [41]. Poor claw health has been linked to an increased prevalence of clinical mastitis [41]. According to our study, we found that blood cortisol concentration had the strongest positive correlation with milk electrical conductivity indicators. 

One limitation of our study was the small size of the groups. 64 cows with signs of lameness and 56 healthy cows were selected. Following research, the number of cows should be increased. Furthermore, in a future study, factors such as heat stress, estrus, and other influences that influence rumination behavior biomarkers should be investigated.

As a result, to the best of our knowledge, this is the first study to investigate the relationship between milk electrical conductivity biomarkers and lameness in dairy cows.

## 5. Conclusions

Based on the hypothesis of our study to find an association of electrical conductivity variables of milk flow with lameness in dairy cows, we found that the average cortisol concentration in the blood is associated with the laminitis score and milk electrical conductivity indicators. Cows with a higher score of lameness had higher cortisol concentrations and higher milk conductivity. We recommend that this investigation be repeated using current methodology in a large number of dairy herds because we could not draw any broad conclusions about the cause and effects because we only investigated the correlation between lameness and milk conductivity in one herd. 

## Figures and Tables

**Figure 1 vetsci-10-00047-f001:**
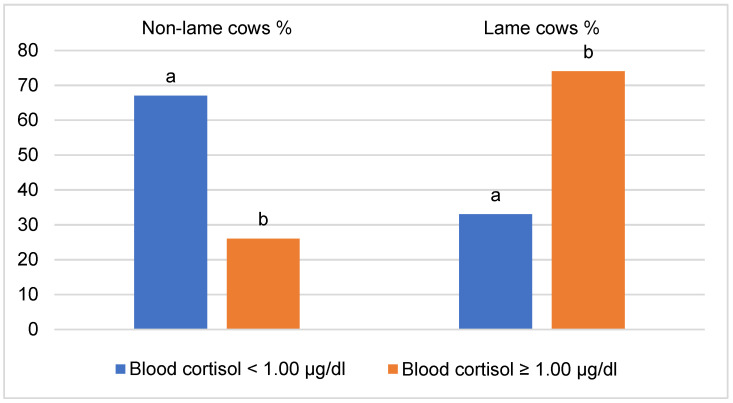
Distribution of non-lame and lame cows by blood cortisol level. ^a, b^—different letters indicate that the difference in the frequencies of the groups in terms of blood cortisol level is statistically significant (*p* < 0.05).

**Figure 2 vetsci-10-00047-f002:**
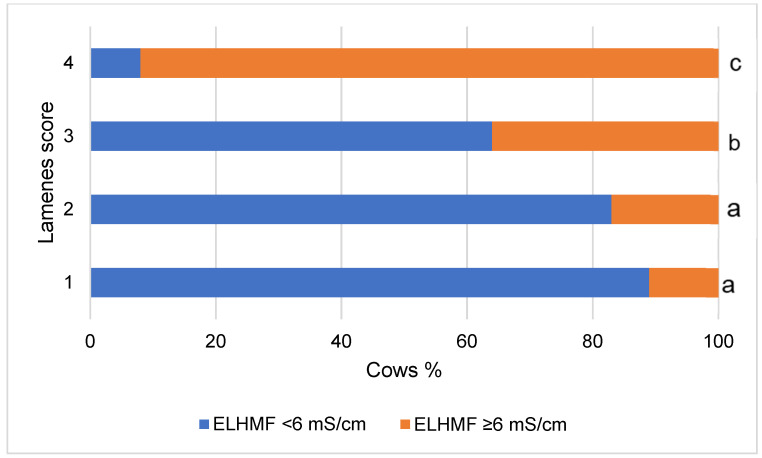
Relationship of lameness intensity with ELHMF level. ELHMF—Electrical conductivity at the maximum milk flow. ^a, b, c^—different letters indicate that the difference in the frequencies of the ELHMF groups in terms of lameness score is statistically significant (*p* < 0.05).

**Figure 3 vetsci-10-00047-f003:**
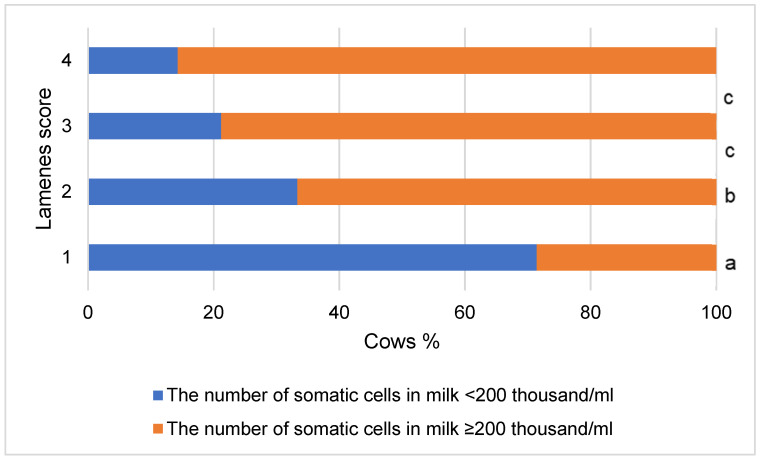
Relationship of lameness intensity with the level of somatic cells in milk. Lameness level—lameness score (1 = normal or non-lame cows, 2 = presence of a slightly asymmetric gait, 3 = the cow clearly favored one or more limbs (moderately lame), 4 = severely lame). ^a, b, c^—different letters indicate that the difference in the frequencies of the somatic cell groups in terms of the lameness score is statistically significant (*p* < 0.05).

**Table 1 vetsci-10-00047-t001:** A detailed description of the electrical conductivity characteristic.

Indicator	Description
**ELHMF**	Electrical conductivity at the maximum milk flow (mS/cm)
ELAP	Electrical conductivity during the first few minutes of milking (mS/cm) (beginning peak level of the electrical conductivity)
ELMAX	Maximum electrical conductivity after reaching the highest milking speed (mS/cm)
ELMNG	Maximum electrical conductivity following main milking (mS/cm)
ELAD	Beginning of the electrical conductivity peak difference (mS/cm)

**Table 2 vetsci-10-00047-t002:** The percentage of non-lame and lame cows based on milk electrical conductivity levels in different phases of milk flow.

Group of Cows	ELHMF		ELAP		ELMAX		ELMNG		ELAD	
					mS/cm					
	<6	≥6	<6	≥6	<6	≥6	<6	≥6	<6	≥6
Non-lame	90.8	9.2	55.4	44.6	75.4	24.6	92.3	7.7	92.0	8.0
Lame	61.8	38.2	27.3	72.7	50.9	49.1	74.5	25.5	72.5	27.5
χ^2^ statistic	χ^2^ = 14.320, *p* < 0.001		χ^2^ = 9.634, *p* = 0.002		χ^2^ = 7.762, *p* = 0.005		χ^2^ = 14.310, *p* < 0.001		χ^2^ = 14.300, *p* < 0.001	

**Table 3 vetsci-10-00047-t003:** Correlation of blood cortisol concentration and lameness score with electrical conductivity variables of milk flow in 64 cows with signs of lameness and 56 healthy cows (with an average of 2.8 lactations and 60 days of the postpartum period).

Indicator	Cortisol	Lameness Score
Cortisol	-	0.457 *
ELHMF	0.704	0.339 *
ELAP	0.519	0.404 *
ELMAX	0.633	0.264 *
ELMNG	0.081	0.153
ELAD	0.060	0.284 *

**Table 4 vetsci-10-00047-t004:** Analysis of 64 cows with signs of lameness and 56 healthy cows (with an average of 2.8 lactations and 60 days of the postpartum period) revealed all milk electrical conductivity indicators were associated with the occurrence of cow lameness.

Factor	*p*	OR	95% C.I. for OR	
			Lower	Upper
ELHMF (class)	<0.001	6.074	2.233	16.52
Cortisol (class)	<0.001	5.704	2.422	13.431
Constant	0.010	0.576		

## Data Availability

The data presented in this study are available within the article.

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
