# Peer review of "Association between Milk Electrical Conductivity Biomarkers with Lameness in Dairy Cows"

_vetsci, 2023, doi:10.3390/vetsci10010047_

Round 1
Reviewer 1 Report
General comments
This is a study about the correlation between lameness and milk conductivity in one herd. Be careful not to conclude too broad about the cause and effects because you only observe one herd. The benefit of samling more herds and the drawback of only having one herd included should be discussed in the paper.
Specific comments
Line 2 title: change to: Association between mastitis biomarkers with lameness in dairy cows.
L34: change: “a higher risk of mastitis” to “a higher milk conductivity”.
L40: Not all references got such a high prevalence so add a few more references here.
L53: Reference no. 10 must be wrong here since that one got nothing to do with AMS.
L73: Early identification of lameness will do nothing unless cows are treated, so that should be added.
L100-101: Unimportant information, delete.
L111: How were cows fixed during blood sampling and how was the sample collected?
L119: Higgs et al., 2014 is not in the list of references and should be noted with a number instead.
L146-147: Is it possible that the procedure of moving cows to a fixed position and taking the blood sample could have influenced level of cortisol?
L154-155: Table 2 is difficult to read (messy) and df does not appear in the table.
L181: You cannot conclude from this study that the risk of developing lameness increases with a higher conductivity, but you can conclude that there is a significant association.
L261: You cannot conclude about the risk of mastitis but about an association with conductivity in milk.
Author Response
Dear Reviewer,
Authors are very thankful for the comments, which help us to improve the manuscript. All changes proposed have been included in the manuscript and highlighted in yellow and track changes.
Best Regards,
Prof. Ramunas Antanaitis
|
Question |
Answers |
|
General comments This is a study about the correlation between lameness and milk conductivity in one herd. Be careful not to conclude too broad about the cause and effects because you only observe one herd. The benefit of samling more herds and the drawback of only having one herd included should be discussed in the paper
|
We corrected and added information in conclusion section – Cows with a higher score of lameness had a higher cortisol concentration and a higher milk conductivity.
We recommend to repeat this investigation using current methodology in a large number of dairy herds because we couldn't draw any broad conclusions about the cause and effects because we investigated the correlation between lameness and milk conductivity in one herd.
|
|
Specific comments |
|
|
Line 2 title: change to: Association between mastitis biomarkers with lameness in dairy cows. |
We corrected title to – “Association Between Milk Electrical Conductivity Biomarkers With Lameness In Dairy Cows” |
|
L34: change: “a higher risk of mastitis” to “a higher milk conductivity”. |
We corrected to “a higher milk conductivity” in introduction and conclusion sections. |
|
L40: Not all references got such a high prevalence so add a few more references here. |
We corrected and added few new references - The prevalence of clinical lameness in cows ranges from 26 to 54% |
|
L53: Reference no. 10 must be wrong here since that one got nothing to do with AMS. |
We corrected and added new reference - The use of AMSs also has the advantage of monitoring cow-level milking frequency, quarter-level production, and milk quality, which can aid in the identification of sickness [14]. |
|
L73: Early identification of lameness will do nothing unless cows are treated, so that should be added. |
We corrected to –“ Early identification and treatment of lameness at all phases of lactation improves milk yield and reduces the incidence of mastitis in the herd” |
|
L100-101: Unimportant information, delete. |
We deleted this information |
|
L111: How were cows fixed during blood sampling and how was the sample collected? |
We corrected to – “During the same general clinical examination, blood samples were taken from the coccygeal vessels of cows fixed with special equipment by...“ |
|
L119: Higgs et al., 2014 is not in the list of references and should be noted with a number instead. |
We corrected and added number of reference – “Data for human and canine T4 and cortisol accuracy and performance, including analyte recovery and dilutional tests, had previously been assessed [23]” |
|
L146-147: Is it possible that the procedure of moving cows to a fixed position and taking the blood sample could have influenced level of cortisol? |
We think that it is not possible. |
|
L154-155: Table 2 is difficult to read (messy) and df does not appear in the table. |
We have corrected the table |
|
L181: You cannot conclude from this study that the risk of developing lameness increases with a higher conductivity, but you can conclude that there is a significant association. |
We corrected to – “Analysis of the final statistical model of the studied influencing factors on lameness in cows showed that there is a significant association between an increases of milk ELHMF ≥6 mS/cm (6.074 times, p < 0.001) and cortisol concentration in blood above 1.00 µg/dl (5.704 times, p < 0.001)” |
|
L261: You cannot conclude about the risk of mastitis but about an association with conductivity in milk.
|
We corrected to – “. Cows with a higher score of lameness had a higher cortisol concentration and a higher milk conductivity” |
Reviewer 2 Report
You run an observational study to assess if blood cortisol, milk electrical conductivity, and locomotion score are associated (L 76-78). The working hypothesis is that milk electrical conductivity is associated with locomotion score (L 74-76).
Most of the introduction section (L 38-74) deals with the negative effects of lameness and, also, with the importance of its early detection (e.g.: L 70-74). So, after reading it, it's expected that the study is focused on assessing risk factors for lameness. Surprisingly, that's not the case. That's confusing!
I have some issues with the way this study was designed.
One concern has to do with the causal diagram behind the objectives of the present study (L 76-78). To determine if blood cortisol and electrical conductivity in milk are associated and if they are related to cow lameness score. This proposal has some potential issues. I mean, cortisol and electrical conductivity are parameters that can change in hours whereas, lameness (e.g., locomotion score) is an event that doesn't change in hours. It's rather a chronic process. Also, cortisol levels, a biomarker of stress, are influenced by other stressors (diseases, heat or cold stress, parturition, etc.) not just by lameness. Milk electrical conductivity is influenced by udder health (e.g., mastitis) but not by lameness. Lame cows show reduced activity and higher lying time leading to an increased risk for mastitis. I mean, the link between lameness and cortisol is clear, the link between mastitis and cortisol is also clear, and the link between mastitis and milk electrical conductivity is clear, but, conversely, the link between electrical conductivity and lameness is not clear. There is an issue with the causal diagram. So, milk electrical conductivity cannot be used as a risk factor for lameness because mastitis is acting as an intermediate variable.
Another concern has to do with the fact that risk factors (L 138-140, L 180-185, and Table 4) are evaluated at the same time as (or maybe later than) lameness is diagnosed through locomotion score. The exposition to risk factors (e.g., cortisol level [low vs. high] or electrical conductivity [low vs. high]) should be before the detection of the event (e.g., lameness) as to be considered as a potential risk factor, and this is not the case in this study!
Additionally, how did you select these cows (L 94-95)? I mean, which are the inclusion criteria used? Also, why both groups are unbalanced (64 vs. 56 cows)? Please, clarify the temporal association between lameness diagnosis and blood sampling for cortisol measurement and milk conductivity evaluation. It is said (L 96-101) that cows were locomotion scored weekly for four weeks, but it's not said exactly when relating to bleeding and conductivity test.
Finally, why did you divide cortisol as <1 vs. >=1 ug/dL, and the same for electrical conductivity as <6 vs. >=6 mS/cm (L 141-144)? Where do you get these cut-offs from?
Minor comments
Title: Please revise as follows "Association of..."
L 22: Please, delete ")" after the postpartum period
Keywords: Add milk to electrical conductivity. I'd replace lameness (that is in the title) by locomotion score. Also, I think that cortisol should be included instead of mastitis (it's also in the title).
L 57: "Antanaitis et al. (2021) [16]" is number 28 in the list of references. Please, make sure that all citations in the text agree with the given numbers in the reference list.
L 72: Please, add "to" after According.
L 79-85: Please define the type of observational study you run (I’d say it’s a transversal study).
Table 3: Add information about the study in the title. I think that this table should be simplified. I mean, the column "statistic" should be removed. The rows with p values should also be deleted. Finally, the r values should be followed by *, **, *** (standing for p<0.1, p<0.05, and p<0.01).
Table 4: Tables are intended to stand alone. So, detailed information about where the data come from is needed in the title (e.g., number and type of cows, farm, etc.).
Explain the abbreviations in the table's footnote (e.g., OR, 95%CI, etc.).
This table contains unneeded information such as values for B, S.E.B, Wald, and df.
Also, in the column named "factor" the levels for ELHMF should be above vs. below the threshold (<6 mS/cm vs. >=6 mS/cm) instead of class vs. constant. The same for cortisol.
Table 5: Please add more information to the title. The table should stand alone. The values for B, SE, Wald, and df are not needed.
This logistic model is assessing the risk of getting a high milk electrical conductivity (mastitis) depending on locomotion score (by one point of increase in locomotion score). So, you are using lameness to predict mastitis, not otherwise.
Author Response
Dear Reviewer,
Authors are very thankful for the comments, which help us to improve the manuscript. All changes proposed have been included in the manuscript and highlighted in yellow and track changes.
Best Regards,
Prof. Ramunas Antanaitis
|
Question |
Answers |
|
You run an observational study to assess if blood cortisol, milk electrical conductivity, and locomotion score are associated (L 76-78). The working hypothesis is that milk electrical conductivity is associated with locomotion score (L 74-76). |
We corrected hypothesis – “According to the literature we hypothesized that there are associations of electrical conductivity variables of milk flow, blood cortisol concentration with lameness in dairy cows” |
|
Most of the introduction section (L 38-74) deals with the negative effects of lameness and, also, with the importance of its early detection (e.g.: L 70-74). So, after reading it, it's expected that the study is focused on assessing risk factors for lameness. Surprisingly, that's not the case. That's confusing! |
In introduction section we added – “Despite the unquestionable importance of lameness (caused by claw horn abnormalities), little is known about the genesis and pathophysiology of the lamenes-associated noninfectious diseases [4] Previous research has found that lameness affects the frequency of visits to the automatic milking system (AMS), cow productivity, and milking intervals. The aggregate of these negative outcomes has a significant influence on herd profitability as well as cow health and wellbeing. It is strongly advised to conduct a full investigation of AMS factors in order to ensure appropriate management of dairy cow performance and hoof health [21]. According to Miguel-Pacheco et al. [22], more research is needed to study the potential use of capabilities and maximum benefits of the technologies accessible in AMS as a tool for measuring and monitoring cow health status. Mazrier et al. (2006)[23]demonstrated that using electronic devices to record cow walking time can detect lameness 7-10 days before clinical indications appear, which is associated with decreased activity in cows. When compared to non-lame cows, cows with lameness spend less time feeding and are less active [24], [25]. Automated lameness detection could fill a knowledge limitation by providing relevant cow and herd information, particularly for mild and moderately lame cows. Early identification and automatic drafting could minimize the time between the start of lameness and treatment, preventing cases from becoming severe, hastening recovery, increasing output, and enhancing welfare [26]. Precision livestock farming is acknowledged as essential for future dairy producers since it allows for constant monitoring of animal health and welfare throughout production [27].
|
|
One concern has to do with the causal diagram behind the objectives of the present study (L 76-78). To determine if blood cortisol and electrical conductivity in milk are associated and if they are related to cow lameness score. This proposal has some potential issues. I mean, cortisol and electrical conductivity are parameters that can change in hours whereas, lameness (e.g., locomotion score) is an event that doesn't change in hours. It's rather a chronic process. Also, cortisol levels, a biomarker of stress, are influenced by other stressors (diseases, heat or cold stress, parturition, etc.) not just by lameness. Milk electrical conductivity is influenced by udder health (e.g., mastitis) but not by lameness. Lame cows show reduced activity and higher lying time leading to an increased risk for mastitis. I mean, the link between lameness and cortisol is clear, the link between mastitis and cortisol is also clear, and the link between mastitis and milk electrical conductivity is clear, but, conversely, the link between electrical conductivity and lamene |
We corrected Lines 76-78: The aim of this study was to determine how lameness in cows correlates with blood cortisol levels and milk electrical conductivity, and which of the indicators of electrical conductivity during milk flow phases can be used to predict early risk of cow lameness.
|
|
Another concern has to do with the fact that risk factors (L 138-140, L 180-185, and Table 4) are evaluated at the same time as (or maybe later than) lameness is diagnosed through locomotion score. The exposition to risk factors (e.g., cortisol level [low vs. high] or electrical conductivity [low vs. high]) should be before the detection of the event (e.g., lameness) as to be considered as a potential risk factor, and this is not the case in this study! |
Our goal was to assess the differences in performance between cows with and without signs of lameness and to identify the most important factors. We believe that data from automated milking and Lactocorder® equipment (ICAR certified and widely used for milking performance) could be used more widely by farmers, consultants and scientists, but more extensive research is certainly needed, your design suggests. We look forward to using your ideas in the future. We believe that our current research may be useful to other researchers and farmers. |
|
Additionally, how did you select these cows (L 94-95)? I mean, which are the inclusion criteria used? Also, why both groups are unbalanced (64 vs. 56 cows)? Please, clarify the temporal association between lameness diagnosis and blood sampling for cortisol measurement and milk conductivity evaluation. It is said (L 96-101) that cows were locomotion scored weekly for four weeks, but it's not said exactly when relating to bleeding and conductivity test. |
About 95 Lithuanian black and white dairy cows that matched the selection criteria were identified. The inclusion criteria were cows that had two or more lactations. |
|
Finally, why did you divide cortisol as <1 vs. >=1 ug/dL, and the same for electrical conductivity as <6 vs. >=6 mS/cm (L 141-144)? Where do you get these cut-offs from? |
According to the electrical conductivity of milk, cows were divided into two groups: <6 mS/cm and ≥6 mS/cm, blood cortisol: <1.00 µg/dl and ≥1.00 µg/dl. These regression model explanatory variables were divided into two categorical classes on the sample mode. |
|
Minor comments |
|
|
Title: Please revise as follows "Association of..." |
We corrected – “Association Between Mastitis Biomarkers With Lameness In Dairy Cows“ |
|
L 22: Please, delete ")" after the postpartum period |
Corrected to –“ ..postpartum period” |
|
Keywords: Add milk to electrical conductivity. I'd replace lameness (that is in the title) by locomotion score. Also, I think that cortisol should be included instead of mastitis (it's also in the title). |
We corrected to – “Keywords: milk to electrical conductivity; locomotion score; cortisol; precision dairy farming” and title – “Association Between Milk Electrical Conductivity Biomarkers With Lameness In Dairy Cows” |
|
L 57: "Antanaitis et al. (2021) [16]" is number 28 in the list of references. Please, make sure that all citations in the text agree with the given numbers in the reference list. |
Corrected |
|
L 72: Please, add "to" after According. |
Corrected to – “According to Garvey…” |
|
L 79-85: Please define the type of observational study you run (I’d say it’s a transversal study). |
We added information – “This transversal study was done at one Lithuanian dairy farm” |
|
Table 3: Add information about the study in the title. I think that this table should be simplified. I mean, the column "statistic" should be removed. The rows with p values should also be deleted. Finally, the r values should be followed by *, **, *** (standing for p<0.1, p<0.05, and p<0.01). |
We corrected title of table 2 – “Table 3.Correlation of blood cortisol concentration and lameness score with electrical conductivity variables of milk flow in 64 cows with signs of lameness and 56 healthy cows (with an average of 2.8 lactations and 60 days of the postpartum period)” We corrected table 2 |
|
Table 4: Tables are intended to stand alone. So, detailed information about where the data come from is needed in the title (e.g., number and type of cows, farm, etc.). |
We added new title – “Analysis of 64 cows with signs of lameness and 56 healthy cows (with an average of 2.8 lactations and 60 days of the postpartum period) revealed all milk electrical conductivity indicators were associated with the occurrence of cow lameness”
|
|
Explain the abbreviations in the table's footnote (e.g., OR, 95%CI, etc.). |
We added information – “OR - odds ratio, CI—95% confidence interval” |
|
This table contains unneeded information such as values for B, S.E.B, Wald, and df. |
We deleted “B, S.E.B, Wald, and df” from this table. |
|
Also, in the column named "factor" the levels for ELHMF should be above vs. below the threshold (<6 mS/cm vs. >=6 mS/cm) instead of class vs. constant. The same for cortisol. |
We've fixed the errors we found |
|
Table 5: Please add more information to the title. The table should stand alone. The values for B, SE, Wald, and df are not needed. |
We corrected to – “Table 5. Association of lameness intensity with electrical conductivity at highest milk flow in 64 cows with signs of lameness and 56 healthy cows (with an average of 2.8 lactations and 60 days of the postpartum period)” We deleted B, SE, Wald and df. |
|
This logistic model is assessing the risk of getting a high milk electrical conductivity (mastitis) depending on locomotion score (by one point of increase in locomotion score). So, you are using lameness to predict mastitis, not otherwise. |
We corrected conclusion section – “Cows with a higher score of lameness had a higher cortisol concentration and a higher milk conductivity” |
Round 2
Reviewer 2 Report
The authors run a transversal observational study to assess if blood cortisol, milk electrical conductivity, and locomotion score are associated, and, also to assess if electrical conductivity can be used to predict the early risk of cow lameness (L 151-154). Despite the changes included in the revised manuscript, the main problems with the study design were neither addressed nor solved. I insist that there is an issue with the causal diagram (concept map). I mean, lameness could lead to mastitis, mastitis causes milk's electrical conductivity alterations, but lameness does not have any direct effect on milk’s electrical conductivity. So, mastitis is acting as an intermediate variable. I mean, lameness is acting indirectly, through mastitis, on milk’s electrical conductivity. Therefore, milk electrical conductivity cannot be used as a risk factor for lameness. Milk’s electrical conductivity is directly influenced by udder health (e.g., mastitis) but not by lameness.
Another concern has to do with the chronological order of exposition to risk factors and disease events. In this study, exposition to risk factors was evaluated after lameness diagnosis (through locomotion score). In L 186-189, it’s said that 64 cows with signs of lameness and 56 healthy cows were included in the study. So, the exposition to risk factors (e.g., cortisol level [low vs. high] or electrical conductivity [low vs. high]) was after lameness cases were identified. That makes no sense because exposition to risk factors should be before the event (e.g., lameness) occurrence.
For the above reasons, I do not recommend this study for publication.
Author Response
Dear Reviewer,
Authors are very thankful for the comments, which help us to improve the manuscript. All changes proposed have been included in the manuscript and highlighted in yellow and track changes.
Best Regards,
Prof. Ramunas Antanaitis
|
Question |
Answers |
|
The authors run a transversal observational study to assess if blood cortisol, milk electrical conductivity, and locomotion score are associated, and, also to assess if electrical conductivity can be used to predict the early risk of cow lameness (L 151-154). Despite the changes included in the revised manuscript, the main problems with the study design were neither addressed nor solved. I insist that there is an issue with the causal diagram (concept map). I mean, lameness could lead to mastitis, mastitis causes milk's electrical conductivity alterations, but lameness does not have any direct effect on milk’s electrical conductivity. So, mastitis is acting as an intermediate variable. I mean, lameness is acting indirectly, through mastitis, on milk’s electrical conductivity. Therefore, milk electrical conductivity cannot be used as a risk factor for lameness. Milk’s electrical conductivity is directly influenced by udder health (e.g., mastitis) but not by lameness.
|
We have updated the article based on your important comments. We supplemented the analysis with an indicator of somatic milk cells as the main biomarker of mastitis, which significantly correlated with the electrical conductivity of milk and laminitis. This confirmed your observations.
|
|
Another concern has to do with the chronological order of exposition to risk factors and disease events. In this study, exposition to risk factors was evaluated after lameness diagnosis (through locomotion score). In L 186-189, it’s said that 64 cows with signs of lameness and 56 healthy cows were included in the study. So, the exposition to risk factors (e.g., cortisol level [low vs. high] or electrical conductivity [low vs. high]) was after lameness cases were identified. That makes no sense because exposition to risk factors should be before the event (e.g., lameness) occurrence. |
We performed a comparative analysis of homogeneous groups. We agree with your comment that it would be appropriate to carry out more extensive research based on your proposed design. We have supplemented the research in this article with additional data. In the group of lame cows, we found 76.56% of milk samples with the level of somatic cells ≥200 thousand/ml, and in the group of non-lame cows, only 28.57% of such samples. (χ2=27.707, p < 0.001). The intensity of lameness was associated with an increase in the number of cows with the indicated level of somatic cells in milk and a decrease in the number of cows in which the level of somatic cells in milk samples was <200 thousand/ml (χ2 = 30.269, p < 0.001) (Figure 3). In the group of cows with ELHMF of milk <6 mS/cm, 36.07% of milk samples with the level of somatic cells ≥200 thousand/ml were detected, and in the group of cows with ELHMF ≥6 mS/cm - even 72.88. % of such samples (χ2=16.374, p<0.001). To analyze the factors associated with lameness in cows, the multivariable logistic regression model (presented in Table 4) was supplemented with somatic cell levels in milk. Applying a backward stepwise logistic regression model, eliminating all non-significant explanatory variables (according to the significance of the Wald criterion), no significant effect of somatic milk cells was found. This can be explained by the fact that highly correlated variables cannot be used in multiple logistic regression models to ensure the absence of multicollinearity, and as mentioned earlier, somatic cell count in milk was significantly associated with ELHMF (p<0.001). |